# The Anti-Tumor Effect of the Newly Developed LAT1 Inhibitor JPH203 in Colorectal Carcinoma, According to a Comprehensive Analysis

**DOI:** 10.3390/cancers15051383

**Published:** 2023-02-22

**Authors:** Rina Otani, Hidehiko Takigawa, Ryo Yuge, Daisuke Shimizu, Misa Ariyoshi, Ryo Miyamoto, Hiroki Kadota, Yuichi Hiyama, Ryohei Hayashi, Yuji Urabe, Akira Ishikawa, Naohide Oue, Yasuhiko Kitadai, Shiro Oka, Shinji Tanaka

**Affiliations:** 1Department of Gastroenterology, Hiroshima University Hospital, Hiroshima 734-8551, Japan; 2Department of Endoscopy, Hiroshima University Hospital, Hiroshima 734-8551, Japan; 3Clinical Research Center in Hiroshima, Hiroshima University Hospital, Hiroshima 734-8551, Japan; 4Department of Gastrointestinal Endoscopy and Medicine, Hiroshima University Hospital, Hiroshima 734-8551, Japan; 5Department of Molecular Pathology, Hiroshima University, Hiroshima 734-8551, Japan; 6Department of Health Sciences, Prefectural University of Hiroshima, Hiroshima 734-8558, Japan

**Keywords:** LAT1, chemotherapy, colorectal cancer, comprehensive analysis, tumor microenvironment

## Abstract

**Simple Summary:**

The large neutral amino acid transporter family (LAT1-4) plays an important role in the intracellular uptake of essential amino acids. Among them, LAT1 is specifically expressed in cancer cells. A novel LAT1-specific inhibitor, JPH203, is expected to cause cancer-specific starvation and possess anti-tumor effects; however, the anti-tumor mechanism for colorectal cancer (CRC) remains unclear. In this study, we clarified the significance of LAT1 expression in cancer progression in database analysis and clinical specimens and demonstrated the utility of JPH203, in vitro and in vivo, using a unique stroma-abundant allogeneic immunoreactive mouse colorectal tumor created by the orthotopic transplantation of a mouse-derived CRC cell line and mesenchymal stem cells into mouse cecum. Comprehensive analysis with RNA sequences showed the efficacy of JPH203 not only in cancer proliferation, but also in tumor stromal activation. These findings will be useful when considering its application in other stroma-abundant carcinomas and its use in combination with other drugs.

**Abstract:**

A novel large neutral amino acid transporter 1 (LAT1)-specific inhibitor, JPH203, is expected to cause cancer-specific starvation and possess anti-tumor effects; however, its anti-tumor mechanism for colorectal cancer (CRC) remains unclear. We analyzed LAT family gene expressions in public databases using UCSC Xena and evaluated LAT1 protein expression using immunohistochemistry in 154 cases of surgically resected CRC. We also evaluated mRNA expression using polymerase chain reaction in 10 CRC cell lines. Furthermore, JPH203 treatment experiments were conducted in vitro and in vivo using an allogeneic immune-responsive mouse model with abundant stroma created via the orthotopic transplantation of the mouse-derived CRC cell line CT26 and mesenchymal stem cells. The treatment experiments were followed by comprehensive gene expression analyses with RNA sequencing. Database analyses and immunohistochemistry research on clinical specimens revealed that LAT1 expression was cancer-dominant, and its increase was accompanied by tumor progression. In vitro, JPH203 was effective in an LAT1 expression-dependent manner. In vivo, JPH203 treatment considerably reduced tumor size and metastasis, and RNA sequencing-based pathway analysis showed that not only tumor growth and amino acid metabolism pathways, but also stromal activation-related pathways were suppressed. The results of the RNA sequencing were validated in the clinical specimens, as well as both in vitro and in vivo. LAT1 expression in CRC plays an important role in tumor progression. JPH203 may inhibit the progression of CRC and tumor stromal activity.

## 1. Introduction

Glucose and amino acid transporters are representative transporters that bring nutrients into cells [1]. Among these, the large neutral amino acid transporter (LAT) plays a critical role in the cellular uptake of amino acids that are essential for human activity.

Recently, it has been reported that the expression levels of the LAT family of proteins are correlated with cancer proliferation via mTOR pathway activation [2], angiogenic activity [3], and clinical prognosis [4] in various carcinomas. LAT1 is strongly correlated with clinical prognosis [5] and tumor-specific upregulation [6]. Furthermore, it reportedly correlates with cisplatin resistance [7] and is attracting attention as a therapeutic target and biomarker for biliary tract cancer and other cancers [8]. LAT1 is expressed specifically in tumors and can be referred to as a “tumor cell type transporter”, whereas LAT2 is mainly expressed in normal cells and can be referred to as “a normal cell type transporter.” Since pan-LAT inhibitors also inhibit LAT2 simultaneously, their toxicity to normal cells has been reported [9]. The association between LAT1 expression and prognostic factors in colorectal tumors [10] and its association with proliferative potential in vitro using colorectal cancer (CRC) cell lines have been previously reported; however, the specific corresponding mechanisms of which have not yet been clarified [11].

LAT1 inhibitors are unique agents that inhibit the growth of cancer cells. The pan-LAT inhibitor 2-aminobicyclo-(2,2,1)-heptane-2-carboxylic acid (BCH), which broadly inhibits LAT1-4, has mainly been used in previous reports, and its tumor growth inhibitory activity has been reported in vitro and in vivo in therapeutic experiments [12,13]. However, because BCH also inhibits LAT family proteins other than LAT1, it is expected that LAT1 inhibitors will be superior to BCH in terms of fewer side effects.

Recently, JPH203, a novel LAT1-specific inhibitor, was developed and reported to inhibit tumor growth [14,15]. The high selectivity of JPH203 for LAT1 compared to that for BCH [15] suggests that JPH203 is an effective drug with fewer side effects because it has less of an effect on normal tissues. Only one report directly compared BCH and JPH203 [15]. This demonstrated the superiority of JPH203 by comparing both drugs in an in vitro experiment using an osteosarcoma cell line. Inhibition of LAT1, which has been reported to show tumor-dominant expression and an association with prognosis, is expected to suppress tumor progression by reducing tumor-specific amino acid uptake, leading to starvation, with minimal effect on normal cells.

In this study, we aimed to elucidate the relationship between LAT1 expression and tumor progression and to investigate the anti-tumor effects and mechanism of LAT1 inhibition in CRC using clinical samples, in vitro and in vivo experiments, and comprehensive analyses of gene expression profiles.

## 2. Materials and Methods

### 2.1. LAT Expression Based on UCSC Cancer Genomics Browser Analysis

UCSC Xena (https://xena.ucsc.edu/ (accessed on 21 May 2020)) is a statistical mining tool used to explore functional genomic datasets for correlations between genomic and phenotypic variables [16]. Using this platform, LAT1 expression in normal tissue and tumor tissues obtained from the colon and other organs was analyzed and evaluated in relation to the prognosis. The relative expression levels of LAT1 and LAT2 in the normal colorectal mucosa and colorectal tumors were also evaluated. 

### 2.2. Patients

In total, 154 consecutive patients with colon cancer who underwent surgical resection for CRC between 2013 and 2015 at Hiroshima University Hospital were enrolled. The study was conducted in accordance with the Declaration of Helsinki and approved by the Institutional Review Board of Hiroshima University Hospital (approval number [E2020-2051]). Although the Ethics Committee of Hiroshima University Hospital waived the requirement for informed consent because we used anonymized data, informed consent was obtained using an opt-out option.

### 2.3. Immunohistochemistry for LAT1 Expression in Human CRC Tissue

To examine the proportion of LAT1 expression and its association with clinicopathological features, LAT1 immunostaining was performed in 154 surgically resected CRC specimens. Formalin-fixed, paraffin-embedded tumor tissues were cut into serial 4 μm sections and examined for LAT1 expression using immunohistochemistry. Staining intensity scores for LAT1 expression were evaluated as follows: 0, no staining; 1+, weak; 2+, moderate; 3+, strong reaction intensity.

### 2.4. Classification of Human Colon Cancer Tissue According to Microsatellite Status

To examine the relationship between LAT1 efficacy and the microsatellite instability (MSI) status, CRC tissue samples were examined for the MSI status either by immunohistochemical staining for mismatch repair protein expression or by PCR amplification of microsatellite sequences using the resected specimens. MSI determination by PCR was performed using pentaplex PCR as described previously [17].

### 2.5. Evaluation of Human CRC Stromal Volume

Formalin-fixed, paraffin-embedded tumor tissues were cut into serial 4 μm sections and immunostained with anti-vimentin antibodies to evaluate stromal volume in human CRC. After vimentin immunostaining, the proportion of vimentin-positive areas in the CRC was evaluated. The correlation between the number of vimentin-positive regions and LAT1 expression in CRC was also examined. Observations were made using a BZ-X710 all-in-one fluorescence microscope (KEYENCE, Osaka, Japan). Microscopic fields that included the foci were photographed at 400× magnification for each specimen and analyzed. All the micrographs were obtained under the same conditions (exposure time, gain, illumination light intensity, and aperture stop). The vimentin-positive areas were determined by arranging and quantifying the brightness thresholds using the BZ-H3C hybrid cell count application of the BZ-X analysis software, version 1.3.1.1 (KEYENCE).

### 2.6. RNA Extraction and Quantitative PCR to Evaluate LAT1 Expression in CRC Cell Lines

As previously described [18], RNA extraction and quantitative PCR were performed using the RNeasy Kit (Qiagen), first-strand cDNA synthesis kit (Amersham Biosciences, Piscataway, NJ, USA), and LightCycler FastStart DNA Master SYBR Green I Kit (Roche Diagnostics, Basel, Switzerland). Reactions were performed in triplicate. To correct for differences in both RNA quality and quantity between samples, expression values were normalized to those of glyceraldehyde-3-phosphate dehydrogenase. Primer sequences were based on previous reports [19] and are provided in Appendix A. The relative LAT1 expression in each cell line was calculated according to the ΔΔCT method, and the normal human/mouse colon mucosa was used as a control.

### 2.7. Cell Lines

The Balb/c mouse colon cancer cell line CT26 was obtained from the American Type Culture Collection (Manassas, VA, USA). The human colon cancer cell lines DLD1, SW480, HCT116, KM12C, Lovo, HT29, WiDr, and Caco2 were obtained from the Health Science Research Resources Bank (Osaka, Japan), and KM12SM was kindly gifted by Dr. Isaiah J. Fidler (University of Texas, Austin, TX, USA). Balb/c mouse-derived mesenchymal stem cells (MSCs) were obtained from Cyagen Biosciences Inc. (Tokyo, Japan).

### 2.8. JPH203 Effects on Cancer Cell Proliferation In Vitro

Cancer cells were maintained in Ham’s F-12 Nutrient Mix medium (Life Technologies Corporation, Carlsbad, CA, USA) supplemented with 10% fetal bovine serum (Sigma-Aldrich, St. Louis, MO, USA) and penicillin–streptomycin mixture for in vitro experiments. Among human-derived 9 colon cancer cell lines (DLD1, SW480, HCT116, KM12C, Lovo, HT29, WiDr, Caco2, and KM12SM) and the mouse-derived colon cancer cell line (CT26), cell lines showing LAT1-high expression, as well as those showing LAT1-low expression, were used for the proliferation assay to evaluate the effect of JPH203 on LAT1 expression. Colon cancer cell lines (cell density, 6 × 10^4^ cells per well) were seeded into 24-well plates (Essen ImageLock; Essen Bioscience, Ann Arbor, MI, USA). The cells were treated with various concentrations of JPH203. Growth curves were generated from a bright field image obtained using a label-free, high-content time-lapse assay system (Incucyte Zoom; E Essen BioScience, Ann Arbor, MI, USA) that automatically expresses cell confluence, as previously described [20]. All experiments were performed in quadruplicate.

### 2.9. JPH203 Effects in CT26 and MSC Co-Culture System

To distinguish CT26 from MSCs, green fluorescent protein (GFP) and puromycin resistance genes were transfected into CT26 colon cancer cells using copGFP control lentiviral particles (sc-108,084; Santa Cruz Biotechnology (Santa Cruz, CA, USA)) according to the manufacturer’s protocol as previously described [21]. The cells were maintained in a complete medium containing puromycin for two weeks after transduction. In the co-culture experiment, CT26-GFP and MSCs were maintained in RPMI1640 with 10% fetal bovine serum (Sigma-Aldrich, St. Louis, MO, USA), a penicillin–streptomycin mixture, and 1 ng/mL of fibroblast growth factor-2. CT26-GFP cells (cell density, 3 × 10^4^ cells per well) cultured with MSCs (cell density, 3 × 10^4^ cells per well) were seeded into 24-well plates (Essen ImageLock; Essen Bioscience, Ann Arbor, MI, USA). The cells were treated with JPH203 at a 1 μM concentration. The proliferation of CT26-GFP cells and MSCs was measured using a label-free, high-content time-lapse assay system (Incucyte Zoom; Essen BioScience, Ann Arbor, MI, USA). CT26 and MSCs proliferation abilities were calculated separately according to the difference in color.

### 2.10. Treatment Experiment Using JPH203 in an Orthotopic Implantation Tumor Model

Female Balb/c mice were obtained from Charles River Japan (Tokyo, Japan). Animal experiments were approved by the Committee on Animal Experimentation at Hiroshima University (Approval number A20-26). Animal experiments were performed using the Balb/c mouse-derived CT26 CRC cell line with high LAT1 expression. As previously reported, CT26 alone transplantation forms stromal-poor tumors [22]. Thus, we conducted animal experiments using a co-implantation model with cancer cells and MSCs in this study. MSCs are known to migrate to tumor sites and differentiate into carcinoma-associated fibroblasts (CAFs) upon interaction with cancer cells to form stromal-rich tumors [21]. To reproduce the characteristics of stroma-rich colorectal tumors, 5 × 10^4^ CT26 cells (stably expressing firefly luciferase) and 5× 10^5^ Balb/c mouse-derived mesenchymal MSCs in 50 μL of Hank’s balanced salt solution were co-transplanted into the cecal wall of mice to generate an allogeneic immune response and a stroma-rich CRC mouse model for treatment with JPH203. Starting 1 d after transplantation, mice were divided into two groups; one group received 1200 mg/kg/day sulfobutyletherβ-cyclodextrin (SBECD) intraperitoneal injection (control group), and the other group received 50 mg/kg/day JPH203 intraperitoneal injection (treatment group). Orthotopically implanted tumor growth (CT-26/Luc cells) was monitored using bioluminescence imaging (BLI). BLI was carried out noninvasively on days 7 and 14 after treatment using a Lumina II in vivo imaging system (Xenogen, Alameda, CA, USA). After anesthetizing the mice with intraperitoneal anesthesia, 200 µL of D-luciferin was injected intraperitoneally into the mice (150 mg/kg), and images were acquired noninvasively [23]. The experiment was continued for 14 days. On day 15, mice were euthanized and necropsied.

### 2.11. Necropsy Procedures and Histological Evaluation

Mice with orthotopic tumors were euthanized, and their body weights were measured. Following necropsy, tumors were excised and weighed. Tumor volume was calculated as V = (W^2^ × L)/2 (V: volume, W: short diameter, L: long diameter). Regional (celiac and para-aortic) macroscopically enlarged lymph nodes were harvested, and histological analysis was performed to verify tumor metastasis. Tumor tissues were fixed with formalin-free zinc fixative (BD Pharmingen; BD Biosciences, CA, USA), paraffin-embedded, cut into serial 4 mm sections, and then examined by pathological and immunohistochemical evaluation.

### 2.12. Double Immunofluorescence for E-Cadherin and Vimentin

Double immunofluorescence staining for E-cadherin and vimentin was conducted as previously reported [21] using the Opal 4-color manual immunohistochemistry (IHC) kit (NEL810001KT; PerkinElmer). Observations were made using a BZ-X710 all-in-one fluorescence microscope (KEYENCE). Representative images were obtained from microscopic fields at 40×, 100×, and 200× magnification for each specimen.

### 2.13. RNA Sequencing and Gene Set Enrichment Analysis (GSEA)

Orthotopically implanted tumors treated with JPH203 or SBECD were dissected and mechanically disrupted using a homogenizer. Total RNA was extracted from tissue homogenates using the Qiagen RNeasy Mini Kit (Qiagen, Hilden, Germany) according to the manufacturer’s protocol. Library construction and data processing were performed at Beijing Genome Institute (Beijing, China). The library was sequenced using the DNBSEQ-G400RS platform, and high-quality reads were obtained. Sequence alignment was conducted using the GRCm38 mouse reference genome version GCF_000001635.26_GRCm38.p6 (https://www.ncbi.nlm.nih.gov/assembly/GCF_000001635.26 (accessed on 10 March 2022)). The HOM_Mouse Human Sequence, downloaded from the Mouse Genome Informatics website (http://www.informatics.jax.org/ (accessed on 19 May 2022)), was used to convert the mouse genes to human genes. After removing genes with FPKM = 0 from all samples, GSEA was performed as previously described [24] to analyze the differential modulation of molecular pathways.

### 2.14. Functional Enrichment Analysis

The differentially expressed gene (DEG) analyses of molecular pathways and gene ontology (GO) terms for cellular components were performed via Dr. Tom’s web according to the path annotation classification from the Kyoto Encyclopedia of Genes and Genomes (KEGG) database [25] and GO annotation classification, and the Phyper function in the R software was used for enrichment analysis. Upon calculating the *p*-value and false discovery rate correcting the *p*-value, the q-value was obtained, and a q-value of <0.05 was regarded as a significant enrichment.

### 2.15. Reagents

JPH203 and SBECD were kindly provided by J-Pharma (Kanagwa, Japan). The primary antibodies used were as follows: monoclonal rabbit SLC7A5/LAT1 antibody NB100-734 from Novus Biologicals (Littleton, CO, USA), polyclonal rabbit E-cadherin antibody from Santa Cruz Biotechnology (Santa Cruz, CA, USA), and vimentin affinity-purified goat polyclonal antibody from Santa Cruz Biotechnology (Santa Cruz, CA, USA).

### 2.16. Statistical Analysis

Clinicopathological features were analyzed using the χ^2^ test or Fisher’s exact test to compare categorical data and Student’s *t*-test or Welch’s *t*-test to compare continuous data. In multiple comparisons for continuous data, one-way ANOVA was conducted. In the clinical data analyses, a post hoc power analysis was conducted to determine the power for the sample size. Spearman’s rank correlation coefficient was used to index correlations. A log-rank test was used to compare the Kaplan–Meier curves and analyze overall survival. Statistical significance was set at *p* < 0.05. All statistical analyses were conducted using EZR software (Saitama Medical Center, Jichi Medical University, Saitama, Japan) [26].

## 3. Results

### 3.1. LAT1 Expression in Humans According to Database Analysis Using Xena

In the database (GTEX and TCGA) analysis using the UCSC Xena platform, LAT1 expression in tumor tissues was markedly higher than that in normal tissues (Figure 1a). Kaplan–Meier survival curves based on LAT1 expression in overall cancer showed that patients with high LAT1 expression had considerably poorer prognoses (Figure 1b). LAT1 expression was higher in metastatic tumors, recurrent tumors, primary tumors, and normal tissues, in this order (Figure 1c). The comparison of LAT1 expression according to cancer type revealed moderate-to-high LAT1 expression in CRC (Figure 1d). Subsequently, an additional analysis was performed, focusing on LAT family expression in CRC. To confirm the specificity of LAT family gene expression in tumor tissues, we analyzed the expression levels of LAT family genes in normal and CRC tissues and found that the expression levels of LAT1 to LAT4 were similar in normal tissues, while LAT1 expression was clearly higher than that of LAT2, LAT3, and LAT4 in CRC tissues. This result indicates that LAT1 expression, but not LAT2, LAT3, and LAT4 expression, is dominant in CRC tissues (Figure 1e). The Kaplan–Meier survival curves for CRC also showed that CRCs with higher LAT1 expression had a poor prognosis (*p* < 0.05) (Figure 1f). In addition, a comparison of LAT1 expression in normal tissues and primary tumors in colon tissues showed that LAT1 expression was higher in primary tumors than in normal tissues (Figure 1g).

### 3.2. Immunostaining in Human CRC Specimens

Among 154 clinical CRC cases, LAT1 was positive in 68.8% of the cases, based on the scale shown in Figure 1h. LAT1 expression was evaluated by classifying LAT1 staining as 0, 1+, 2+, or 3+, with 0 and 1+ defined as negative and 2+ and 3+ as positive. LAT1 was positive in 68.8% of the cases. Post hoc power analysis of data from the 154 patients demonstrated a statistical power greater than 90% for LAT1-positive expression versus LAT1-negative expression (power = 0.99) for both subjective and objective evaluations. When comparing LAT1-positive and LAT1-negative patients, stage, T-classification, and N-classification were extensively higher in the LAT1-positive group. Other clinicopathological characteristics, such as sex, tumor location, histological type, budding grade, MSI, KRAS mutation, and BRAF mutation, were not significantly different (Table 1).

### 3.3. LAT1 Expression in Colon Cancer Cell Lines

In human-derived 9 colon cancer cell lines (DLD1, SW480, HCT116, KM12C, Lovo, HT29, WiDr, Caco2, and KM12SM) and a Balb/c mouse-derived colon cancer cell line (CT26), LAT1 mRNA expression was high in HT29 and CT26 (relative expression: 17.8 and 22, respectively), intermediate in SW480 and HCT116, and low in DLD1 and Lovo (relative expression: 0.3 and 1.5, respectively). There was no significant difference in the LAT1 average expression between the MSS and MSI cell lines (Figure 2a).

### 3.4. Proliferation Assay Using Colon Cancer Cell Lines with JPH203 In Vitro

Using DLD1, Lovo, and SW480 as LAT1-low expression cell lines and HCT116, HT29, and CT26 as LAT1-high expression cell lines, we conducted in vitro experiments to evaluate the effect of JPH203. Each experiment was conducted with various concentrations of JPH203 in the range of 0–20 µM. The evaluation was performed by labeling the area of the cancer cell-occupying region and measuring the area using Incucyte Zoom (Ver. 2018A). As a representative example, the results for CT26 cells (Figure 2b) are demonstrated. All other cell lines were evaluated in the same manner. The results showed that JPH203 inhibited cell proliferation in a concentration-dependent manner, as shown in Figure 2c. This effect was pronounced when LAT1 expression was above a certain level.

### 3.5. JPH203 Administration Effect in a Congenic Mouse Tumor Model with Abundant Stroma

A representative image of the tumors removed 14 days after transplantation is shown in Figure 3a. Luciferase activity in the tumors over time was measured and compared between the control and treatment groups. Luciferase activity was markedly lower in the treatment group than in the control group at 14 days post-transplantation (*p* < 0.05; Figure 3b). Images taken at sacrifice are shown in Figure 3c. Each group comprised 12 mice; however, one mouse in the treatment group showed no tumor incidence. Therefore, collectively, 12 tumors in the control group and 11 tumors in the treatment group were evaluated. The treatment group with JPH203 had an obviously reduced tumor size compared with that in the control group. A summary of the results from the 12 control and 11 treated animals is shown. Marked suppression of tumor weight, volume, and metastasis was observed in the treatment group (Table 2).

### 3.6. RNA Sequence

To analyze the changes in gene expression profiles in mouse tumors induced by JPH203 treatment, RNA was extracted from orthotopically implanted tumors of mice in the control and treatment groups and analyzed using RNA sequencing. The significant DEGs detected were statistically plotted according to the gene expression level of each sample (Figure 4a). JPH203 treatment upregulated 163 genes and downregulated 442 genes. A bubble chart of enrichment using the KEGG PATHWAY analysis is shown (Figure 4b). The pathways with the highest variability were identified and classified. Remarkably, stroma-related pathways were greatly suppressed. Cancer growth pathways and amino acid metabolic pathways were also suppressed. In contrast, glutamine metabolism, which is a non-essential amino acid pathway, was activated. The results of the GO term analysis of the cellular components are shown in Figure 4c. The expression of genes related to the extracellular matrix was decreased. Similarly, in GSEA, the gene sets related to stromal responses were reduced. Proliferation-related gene sets were suppressed, consistent with previous reports. Amino acid-related gene sets were also suppressed, but the suppression of the stromal response was more remarkable (Figure 4d). 

The results of RNA sequence analysis were validated in human specimens, mouse tumor tissues, and in vitro experiments. To validate the effect of tumor stromal suppression by JPH203 on RNA sequencing, immunohistochemical evaluation was performed on the mouse tumors. Evaluation with H&E staining of excised mouse tumors in the SBECD (control) group showed that the tumor exhibited infiltrative invasion and abundant stromal reaction. The co-implantation of CT26 and MSCs produced stroma-abundant tumors, as we expected. The tissue structure consisted of a mixed component of the epithelial phenotype area and mesenchymal phenotype with a spindle-shaped morphology area (Figure 5a). In the group treated with JPH203, tumors developed expansively with poor stroma. The JPH203-treated group showed an epithelial phenotype and a homogeneous tissue structure with poor heterogeneity (Figure 5b). The control group showed an abundant area positive for vimentin staining, a representative stromal marker, in the fluorescent double immunostaining of mouse tumors. Merged regions of E-cadherin and vimentin expression, indicating epithelial and mesenchymal transition (EMT), were broadly observed. In contrast, the JPH203 treatment group showed a poor stromal response, and the majority of tumors had an epithelial phenotype with dominant E-cadherin expression (Figure 5c left panels). The vimentin/DPAI-positive area rate was significantly lower in the JPH203 group than in the treatment group. The vimentin/E-cadherin-positive area rate was significantly lower in the JPH203 group than in the treatment group (Figure 5d). We subsequently performed immunostaining for vimentin for validation in 154 human CRC tissue specimens and evaluated the stromal regions and quantified the occupied area, as shown in Figure 5e. Scatter plots for the LAT1 scores and vimentin area (area, %) are shown in Figure 5f. The LAT1 score was evaluated by immunostaining from 0 to 3+, according to the definition described in the Methods section. A correlation was observed between the LAT1 score and vimentin area, with higher LAT1 expression scores indicating higher vimentin expression. The correlation coefficient was 0.714, indicating a strong positive correlation. In vitro, JPH203 inhibited not only CT26 proliferation, but also MSC proliferation in an assay in which cancer cells (CT26) and MSCs were co-cultured (Figure 5g).

## 4. Discussion

In this study, we aimed to elucidate the relationship between LAT1 expression and tumor progression and investigate the anti-tumor effects and mechanism of LAT1 inhibition in CRC. LAT1 expression was higher in colorectal tissues than in normal tissues. LAT1 expression increases with tumor progression, and patients with abundant LAT1 expression have a poor prognosis. Immunostaining of CRC specimens revealed that 68.8% of cases showed LAT1 positive expression, which was dominant at the tumor site. In CRC cell lines, LAT1 was also highly expressed in most cell lines, compared to that in the normal mucosa. JPH203 effectively inhibited the proliferation of CRC cell lines and inhibited tumor growth and metastasis in a mouse model. Comprehensive analysis revealed the suppression of not only proliferative and amino acid metabolic pathways, but also stroma-related pathways upon the administration of an LAT1 inhibitor.

Data-based analysis using UCSC Xena showed predominant expression of LAT1 compared to that of other LAT family members in various cancer types, and a similar trend was observed in CRC. These results were consistent with a previous report, which also found a correlation between LAT1 expression and CRC prognosis [10]. The expression of LAT1 was higher in the following order: metastases, recurrent tumors, primary tumors, and normal tissues. This result suggests that it is a molecule involved in tumor progression and metastasis. It has been previously reported that LAT1 expression correlates with metastasis in CRC [10], and it is interesting to note that LAT1 is also highly expressed in recurrent tumors. In our study of clinical specimens, a similar correlation between LAT1 expression and tumor stage was confirmed. LAT1 is predominantly expressed in tumors of various carcinomas, including CRC [27]. In contrast, LAT2 is expressed in normal tissues, such as renal epithelial cells, brain capillary endothelial cells, neuroendocrine cells [28], and the proximal tubules of the kidney [29]; however, few reports have evaluated LAT2 expression in human tissues. In our database analysis, only LAT1 expression was tumor-dominant, and LAT2, LAT3, and LAT4 were similarly expressed in tumor and non-tumor tissues. These results support the advantage of LAT1 specific inhibition.

We evaluated LAT1 expression in multiple CRC cell lines. In most cell lines (9 of the 10 CRC cell lines), LAT1 expression was higher in cancer cell lines than in normal tissues. This suggests that LAT1 may serve as a potential therapeutic target. These results were almost consistent with a previous report that showed higher LAT1 expression in various cancer cell lines, such as HT29, than in normal tissue [30]. We assumed that the effect of JPH203 was dependent on LAT1 expression; however, JPH203 was effective when LAT1 expression was above a certain level. The fact that JPH203 is a competitive inhibitor [14] may be a contributing factor to this result. In the in vitro experiment using various cell lines, there was no clear difference in LAT1 expression or the effect of JPH203 depending on the MSI status. Similarly, there was no relationship between LAT1 expression and the MSI status in immunostaining studies of clinical specimens. Although the MSS status has been reported to be associated with resistance to immune checkpoint inhibitors [31] and anticancer drugs such as 5-flurouracil [32], no clear relationship between the MSI status and the effects of LAT1 inhibitors was observed in this research.

The IC50 of JPH230 in the CRC cell lines in our experiment was 2 to 10 μM (Figure 2c), although the IC50 was higher than that of general chemotherapeutic agents, such as Paclitaxel (0.0000077 μM) and 5-FU (1.12 μM) [33]. The IC50 of JPH203 for cell proliferation in the CRC cell line HT29 is reported to be 4.1 μM [33], which is similar to our observations. In the previous study, the plasm concentration of JPH203 was only 1000 ng/mL (approximately 2 μM), after administrating a clinically equivalent dose (25 mg/kg) of JPH203. However, the JPH203 administration exhibited 60–80% tumor growth inhibition against mouse tumors under in vivo conditions. These observations suggest that there may be some discrepancy between IC50 in plasma and in vitro conditions in JPH203 usage. The authors discussed that this discrepancy could be attributed to the difference in the substrate concentration (amino acid) between in vitro culture condition and plasma, given that JPH203 inhibits LAT1 by competing with its substrates. Furthermore, as conventional chemotherapeutic agents, such as Paclitaxel and 5-FU, exert cytotoxicity toward not only cancer cells, but also normal cells, untoward effects are unavoidable. In contrast, JPH203, despite its higher IC50, would be expected to exert weaker adverse effects on normal cells because of its higher cancer cell selectivity. These pharmacological properties could redeem the high IC50 of JPH203. In 2023, a phase 2 clinical study also showed effective results at a similar concentration of JPH203 [34]. As the pharmacokinetics of this drug are inferred to be too complicated to be simply explained only based on blood concentrations, the detailed mechanism of pharmacokinetics is an issue to be addressed in future research.

CRC has long been known to have abundant stroma [35]; however, in our previous study, tumors implanted with CT26 alone showed a poor stromal response in an orthoptic-implantation mouse model [22]. We have previously reported that MSCs migrate to tumor sites and differentiate into CAFs through interaction with cancer cells; thus, co-transplantation of MSCs and cancer cells results in the formation of stromal-abundant tumors [18,21]. Furthermore, co-implantation of MSCs with cancer cells by orthotopic implantation leads to the development of tumors with high-metastatic potential as we have previously reported [20,21,22]. Based on these findings, in this study, we used an allogeneic immune-responsive, stromal-abundant, and highly metastatic colon cancer mouse tumor model created by orthotopic transplantation of CT26 and MSCs as a model that reflects the tumor microenvironment to evaluate the effects of drugs on the tumor microenvironment and metastasis.

Histological and gene expression analyses performed in this study showed that JPH203 inhibited tumor stroma, amino acid metabolism, and proliferative pathways in allogeneic CRC lesions. These results are consistent with previous findings that JPH203 suppresses proliferative pathways, such as mTOR [36] and amino acid metabolism [15], which is an intrinsic effect of the drug. Interestingly, while amino acid metabolism was suppressed, the metabolism of non-essential amino acids [15], such as glutamine, was activated, suggesting a compensatory effect for suppressing the essential amino acid pathway. 

In addition, there is a report showing that LAT1 is an essential molecule for the maintenance of the mesenchymal phenotype based on comprehensive analysis in in vitro experiments [15], but there have been few reports analyzing its value in vivo or in clinical specimens, a remarkable finding of this study. Other drugs with tumor-starving effects, such as glutaminase inhibitors and glucose transporter inhibitors, have also been reported to exert inhibitory effects on the stroma [37]. Combining these previous reports and our results, we speculate that tumor starvation therapy may have inhibitory effects on EMT and tumor stromal activation.

It is well known that the interaction between cancer and stroma plays an important role in the growth and metastasis of CRC [38,39]. We focused on the interaction between cancer and stroma in the cancer microenvironment and showed that tumor stroma and stromal activation are associated with EMT in cancer progression, promoting tumor growth and metastasis [21]. It is also shown that inhibiting cancer and tumor stroma is important for inhibiting tumor progression [20,40]. In the clinical setting, the ability of LAT1 to inhibit both tumor cancer cell proliferation and stroma in colorectal tumors, which are considered to have abundant stroma, may be useful in inhibiting tumor progression. When using JPH203 in clinical practice, the possibility of causing side effects on normal stromal cells should be considered. However, immunohistochemistry of the clinical specimens in our study indicated that LAT1 expression on normal stromal cells is relatively low (Figure 1h). It is possible that JPH203 dominantly inhibits tumor stroma with a mild side effect on normal stromal cells. The effect of JPH203 on normal stromal cells should be evaluated in future studies.

In this study, we demonstrated the utility and mechanism of JPH203, a newly developed LAT1-specific inhibitor, to inhibit tumor progression using a unique stroma-abundant allogeneic immunoreactive mouse colorectal tumor created by the orthotopic transplantation of a mouse-derived CRC cell line and MSCs in the mouse cecum. The fact that JPH203 showed inhibitory activity not only on cancer cells, but also on the tumor stroma is considered an important finding and will be useful when considering its application to other stroma-abundant carcinomas and its use in combination with other drugs.

This study has some limitations. First, animal experiments were performed using only one cell line, because there are only a few mouse-derived CRC cell lines that can be used in experiments to evaluate a therapeutic effect under allogeneic immune response conditions. Second, although our results showed that LAT1 inhibitors suppress tumor stroma, the mechanism by which amino acid transporter inhibition suppresses tumor stroma remains unclear and needs to be verified in further studies.

## 5. Conclusions

LAT1 expression in CRC plays an important role in tumor progression. The LAT1 inhibitor JPH203 inhibits not only the proliferative pathway and amino acid metabolism, but also the tumor stromal reaction, and it may therefore serve as an effective novel therapeutic agent.

## Figures and Tables

**Figure 1 cancers-15-01383-f001:**
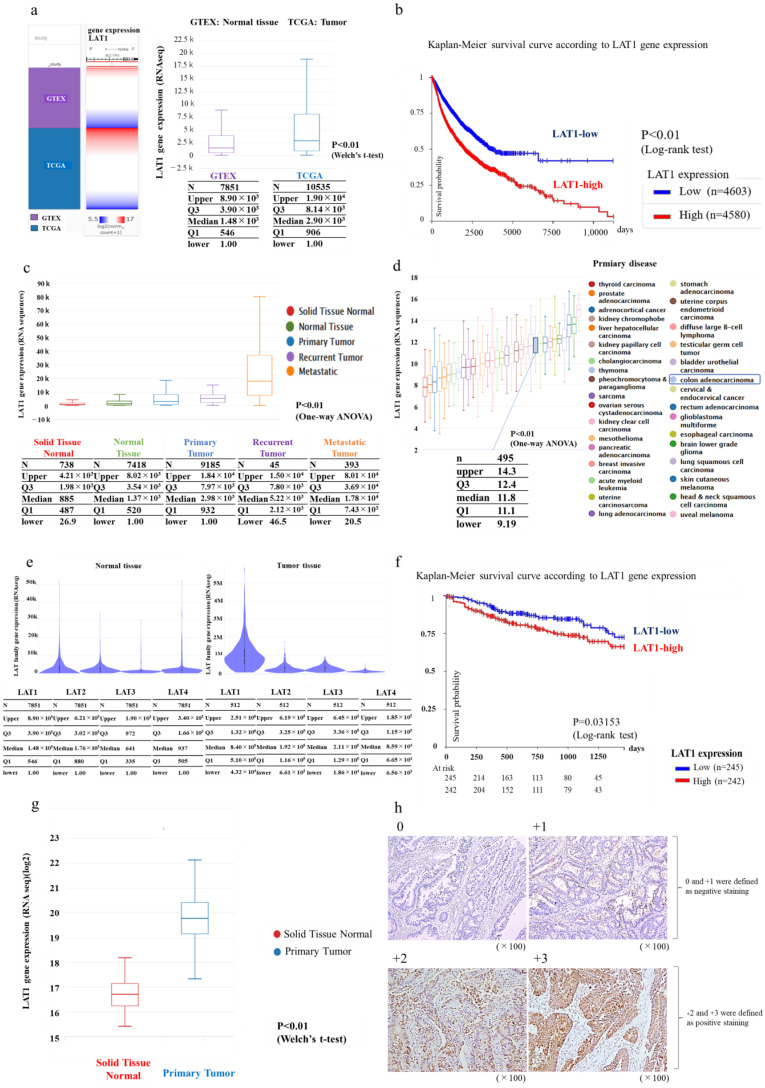
Database analysis based on UCSC cancer genomics browser analysis. (**a**) LAT1 expression in normal and tumor tissues (TCGA TARGET GTEx, 19,131 Samples). Welch’s test was conducted. (**b**) The relationship between LAT1 expression and prognosis in Pan-Cancer (GDC Pan-Cancer, 9372 Samples). The cutoff was set at the median. A log-rank test was conducted. (**c**) LAT1 expression in normal tissue, primary tumor, recurrent tumor, and metastatic tumor (TCGA TARGET GTEx, 19,131 samples). (**d**) LAT1 expression according to the primary site of cancer (TCGA Pan-Cancer, 12,839 samples). (**e**) LAT family expression in normal tissue and colorectal cancer (GTEX, 9783 Samples; TCGA Colon Cancer, 571 samples). (**f**) Relationship between LAT1 expression and prognosis in colorectal cancer (GDC TCGA Colon Cancer (COAD), 571 Samples). The cutoff was set at the median. A log-rank test was conducted. (**g**) LAT1 expression in normal tissue, primary tumor, recurrent tumor, and metastatic tumor (colorectal cancer) (GDC TCGA Colon Cancer (COAD), 571 samples). (**h**) Immunostaining of LAT1 in surgically resected human colorectal cancer specimens.

**Figure 2 cancers-15-01383-f002:**
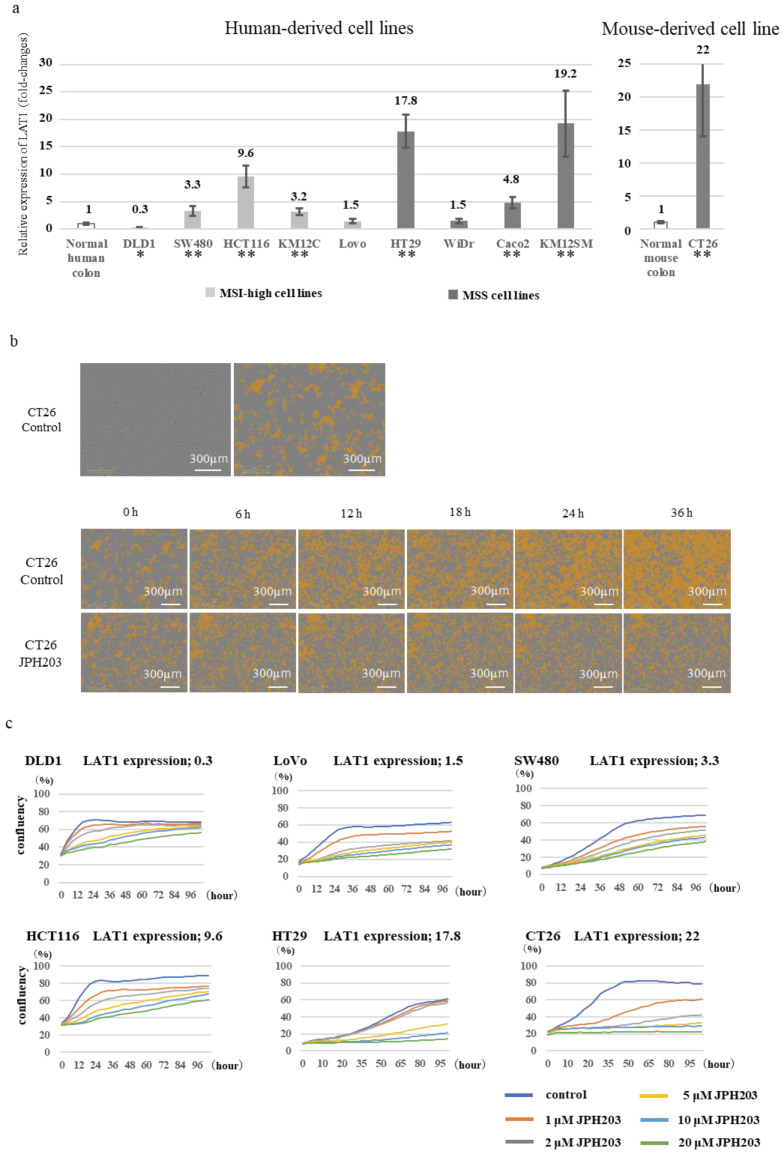
Expression of LAT1 in various cell lines, and the effect of the LAT1 inhibitor JPH203. (**a**) Relative LAT1 mRNA expression in colorectal cancer cell lines. * LAT1 expression was significantly lower compared to that in normal tissue (Student’s *t*-test; *p* < 0.05). ** LAT1 expression was significantly higher compared to that in normal tissue (Student’s *t*-test; *p* < 0.05). (**b**) The area of cells is evaluated upon its measurement using Incucyte. The upper panel shows the representative image of cancer cells labeled by the Incucyte software, whereas the lower panel shows the time-lapse image of CT26 cancer cell proliferation with and without JPH203 exposure (Drug concentration: 1 μM). (**c**) Inhibitory effect on cell proliferation by the LAT1 inhibitor JPH203. DLD1 was significantly more inhibited by JPH203 than by 20 μM (Student’s *t*-test; *p* < 0.05). Lovo, SW480, HCT116, and CT26 were significantly more inhibited by JPH203 than by 1 μM (Student’s *t*-test; *p* < 0.05). HT29 was significantly more inhibited by JPH203 than by 5 μM (Student’s *t*-test; *p* < 0.05).

**Figure 3 cancers-15-01383-f003:**
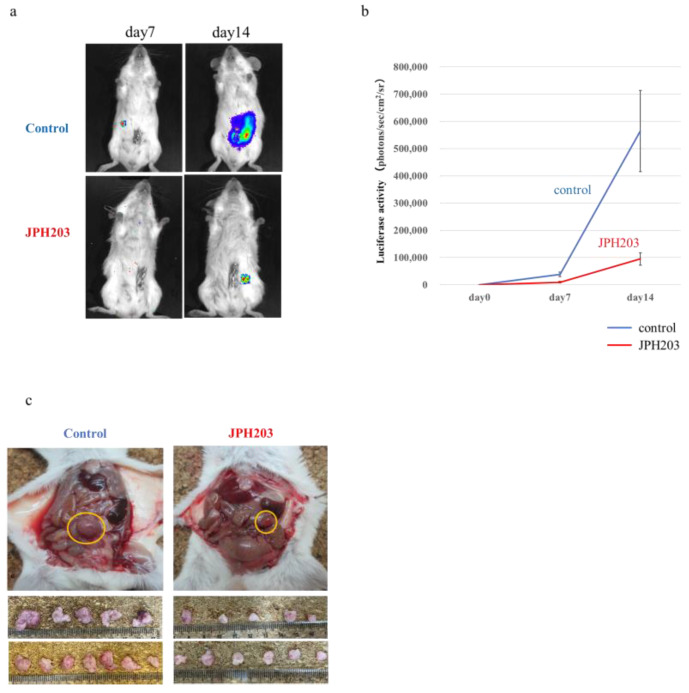
Effect of JPH203 on orthotopic implanted tumor (Control group; *n* = 12 vs. JPH203 treatment group; *n* = 11). (**a**) Representative image of tumor obtained by luminescence observation (Days 7 and 14 post-transplantation). (**b**) Quantification of changes over time according to luminous intensity. Student’s *t*-test was conducted for statistical analysis. (**c**) Macroscopic images of mouse tumors after sacrifice on day 14 post-transplantation.

**Figure 4 cancers-15-01383-f004:**
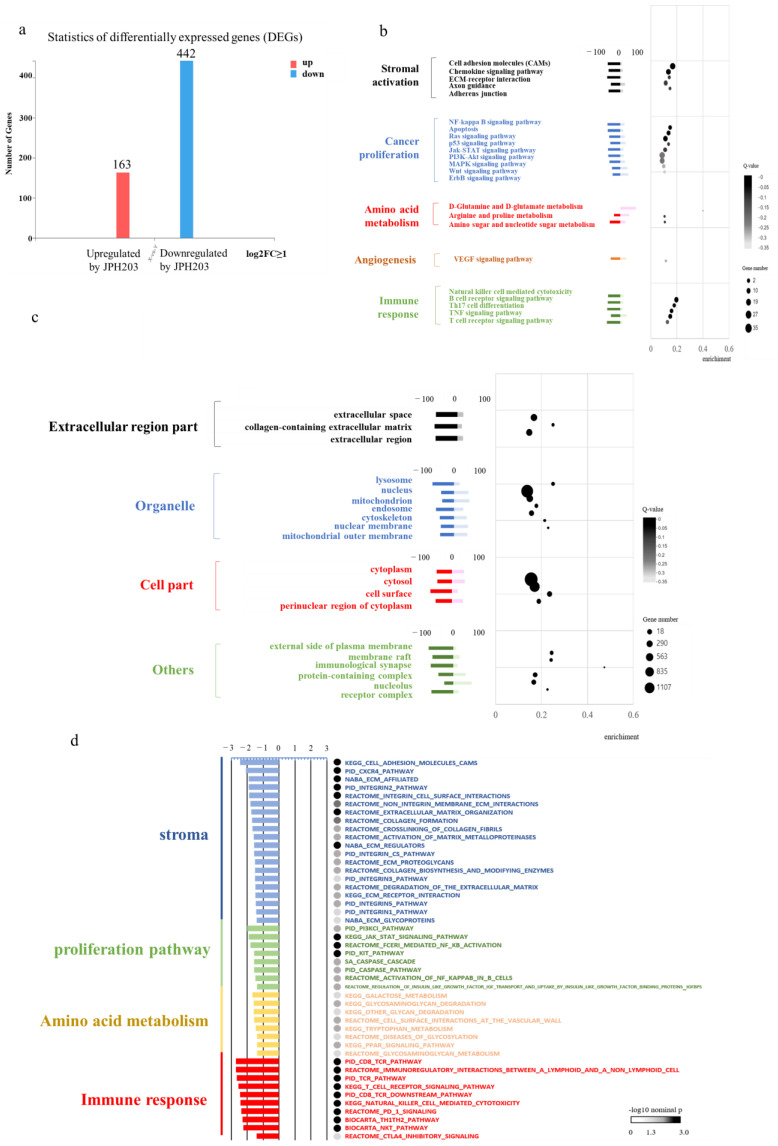
RNA extraction from a CT26 + MSC mouse cecal orthotopic tumor with and without JPH203. (**a**) Statistics of differentially expressed genes (DEGs). According to the gene expression level of each sample, the significant DEGs detected were statistically plotted. (**b**) Bubble chart based on KEGG pathway enrichment analysis. (**c**) GO cellular component enrichment bubble chart. (**d**) Gene set enrichment analysis (GSEA).

**Figure 5 cancers-15-01383-f005:**
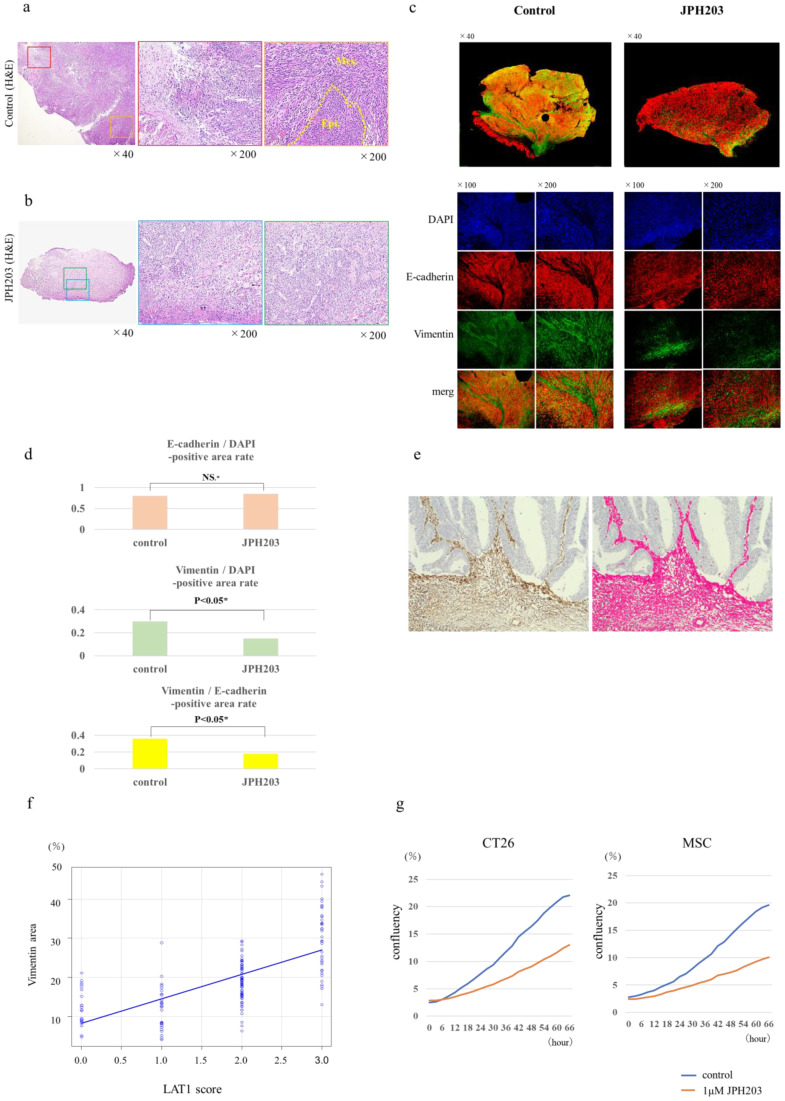
Validation for results of RNA sequences with mouse tumor, human specimens, and in vitro analysis (**a**) Histological analysis of CT26 + MSC mouse cecal orthotopic tumor without JPH203. Control tumor showing infiltrating invasion manner (left and middle panels). Control tumor consists of a mixed component of epithelial phenotype and mesenchymal phenotype (right panel). Epi.; Epithelial phenotype (cobblestone shape), Mes.; Mesenchymal phenotype (Spindle shape). (**b**) Histological analysis of a CT26 + MSC mouse cecal orthotopic tumor with JPH203 (H&E staining). JPH203 showed expanding invasion manner (left and middle panels). Tumor treated with JPH203 consists of homogeneous type component showing epithelial phenotype (right panel). (**c**) Double fluorescence staining using tissue obtained from mouse orthotopic colon tumors. Expression of E-cadherin (red) and vimentin (green) in tumors generated by co-implantation of CT26 cells and MSCs. DAPI nuclear staining is shown in blue. The left panels show the control group, whereas the right panels show the JPH203 treatment group. First line represents merged images (×40), second line represents DAPI image (×100 and ×200), third line represents E-cadherin (red) images (×100 and ×200), fourth line represents vimentin (green) images (×100 and ×200), and fifth line represents merged images (×100 and ×200). (**d**) The quantitative data for double fluorescence staining. The E-cadherin/DAPI-positive area rate (upper graph). The Vimentin/DAPI-positive area rate (middle graph). The vimentin/E-cadherin-positive area rate (lower graph). Student’s *t*-test was conducted for statistical analysis. (**e**) Immunohistochemistry for vimentin in surgically resected human colorectal cancer specimens. The left panel shows the H&E staining image with immunohistochemistry for vimentin. The vimentin-positive areas were determined by arranging and quantifying the brightness thresholds using the BZ-H3C hybrid cell count application of the BZ-X analysis software, version 1.3.1.1 (KEYENCE) (right panel). (**f**) Scatterplot for LAT1 expression and vimentin (Spearman’s rank correlation was conducted to examine the correlation). (**g**) Inhibitory effect on cell proliferation by the LAT1 inhibitor, JPH203. Left panel shows the inhibitory effect of JPH203 on CT26 cells in the co-culture condition of CT26 and MSCs. The right panel shows the inhibitory effect of JPH203 on MSC under co-culture conditions of CT26 and MSCs. Student’s *t*-test was conducted for statistical analysis.

**Table 1 cancers-15-01383-t001:** Association between LAT1 expression and clinicopathological features of human colorectal cancers.

		LAT1 Expression		
		Positive	Negative	*p* *
Number of patients		106 (68.8)	48 (31.2)	
Age (years)		68.8 ± 12.0	66.3 ± 11.9	0.222
Sex	Male	65 (61.3)	26 (54.2)	0.480
Location	Right side colon	34 (32.1)	20 (41.7)	0.277
	Left side colon	72 (67.9)	28 (58.3)	
Histological Type	tub1/2	95 (89.6)	44 (91.7)	0.778
	Por/muc	11 (10.4)	4 (8.3)	
Stage	I/II	46 (43.4)	32 (66.7)	0.009
	III/IV	60 (56.6)	16 (33.3)	
T	1/2	26 (24.5)	22 (45.8)	0.014
	3/4	80 (75.5)	26 (54.2)	
N	N0	48 (45.3)	32 (66.7)	0.015
	N1/2/3	58 (54.7)	16 (33.3)	
M	0	85 (82.3)	42 (79.4)	0.361
	1	21 (17.7)	6 (20.6)	
Budding grade	1	21 (50.0)	9 (37.5)	0.732
	2, 3	21 (50.0)	15 (62.5)	
Microsatellite instability	MSS	98 (92.5)	43 (89.6)	0.545
	MSI-high	8 (7.5)	5 (10.4)	
KRAS mutation	+	42 (39.6)	19 (39.6)	1
BRAF mutation	+	2 (1.9)	4 (8.3)	0.076

Data are represented as *n* (%). * χ^2^ test and Student’s *t*-test were conducted for categorical and continuous data, respectively.

**Table 2 cancers-15-01383-t002:** Results of animal experiments (orthotopic implanted tumor).

Group	Tumor Incident	Body Weight -Before (g)	Body Weight -After (g)	Tumor Weight (g)	Tumor Volume (mm^3^)	Lymph Node Metastasis
Control (*n* = 12)	12/12	21.9 (21.0–23.5)	20.7 (19.0–21.2)	1.15 (0.27–1.58)	312.2 (108–650)	7/12
JPH203 (*n* = 11)	11/11	22.1 (21.5–22.8)	20.4 (18.1–22.6)	0.17 (0.15–0.27)	44.5 (32–108)	1/11
*p*-value *	1.00	0.37	0.49	<0.001	<0.001	0.027

* Fisher’s exact test and Student’s *t*-test were conducted for categorical and continuous data, respectively.

## Data Availability

The data that support the findings of this study are available from the corresponding author upon reasonable request.

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
