# Peer review of "The Anti-Tumor Effect of the Newly Developed LAT1 Inhibitor JPH203 in Colorectal Carcinoma, According to a Comprehensive Analysis"

_cancers, 2023, doi:10.3390/cancers15051383_

Round 1

Reviewer 1 Report

Rina O et. al present a novel inhibitor JPH203 targeting large neutral amino acid transporter 1 (LAT1) on both cancer cells and stroma component in colorectal carcinoma. Combined with publicly available database, they proved the LAT1 highly expressed in CRC patients, which indicated the potential of LAT1 as a onco-target. However, many of the figures could be better labeled or described, and some of the claims may be stronger than is fully justified by the data. There are a number of issues that weaken the paper. Please see the comments and suggestions below:

Major:

The resolution for each figure does not match the 300dpi which is hard to find the information for readers.

What is the IC50 for the JPH203, since 1 uM is already high for the drug development, and most of the cell lines in 1 uM did not reach promising results.

If this inhibitor could target the stroma cells, would that mean the side effect of this inhibitor should be more carefully investigated?

Since the metastasis tumor expressed more LAT1, would that be interesting to see the LAT1 inhibitor effect on controlling metastasis, using a more metastasis prone tumor model.

Minor:

The statistical method should be annotated in each figure legend.      

In figure 2a and 2c, if this difference reaches statistical significance?

In Figure 5c, it would be interesting to show the quantitative information from the image in each channel.

Author Response

[Response]

We are grateful for the positive review of our manuscript and for the constructive comments and useful suggestions.

We have revised the manuscript in response to the reviewers' comments and had our manuscript checked by the English Editing Service Editage (https://www.editage.jp). We have provided our point-by-point responses to the reviewers’ comments below.

Our responses are indicated in red font. Revisions to the manuscript are also indicated in red font.

Rina O et. al present a novel inhibitor JPH203 targeting large neutral amino acid transporter 1 (LAT1) on both cancer cells and stroma component in colorectal carcinoma. Combined with publicly available database, they proved the LAT1 highly expressed in CRC patients, which indicated the potential of LAT1 as a onco-target. However, many of the figures could be better labeled or described, and some of the claims may be stronger than is fully justified by the data. There are a number of issues that weaken the paper. Please see the comments and suggestions below:

Major:

The resolution for each figure does not match the 300dpi which is hard to find the information for readers.

[Response]

We apologize that the low-resolution of our images made them difficult to view. We have created high-resolution images and thoroughly revised all figures to improve the viewability. Thank you for pointing this out.

What is the IC50 for the JPH203, since 1 uM is already high for the drug development, and most of the cell lines in 1 uM did not reach promising results.

[Response]

The IC50 of JPH230 in the CRC cell lines in our experiment was 2 to 10μM (Figure 2c), although the IC50 was higher than that of general chemotherapeutic agents, such as Paclitaxel (0.0000077 μM) and 5-FU (1.12 μM) [33]. The IC50 of JPH203 for cell proliferation in the CRC cell line HT29 is reported to be 4.1 μM [33], which is similar to our observations. In the previous study, the plasm concentration of JPH203 was only 1000 ng/ml (approximately 2 μM), after administrating a clinically equivalent dose (25 mg/kg) of JPH203. However, the JPH203 administration exhibited 60-80% tumor growth inhibition against mouse tumors under in vivo conditions. These observations suggest that there may be some discrepancy between IC50 in plasma and in vitro conditions in JPH203 usage. The authors discussed that this discrepancy could be attributed to the difference in the substrate concentration (amino acid) between in vitro culture condition and plasma, given that JPH203 inhibits LAT1 by competing with its substrates. Furthermore, as conventional chemotherapeutic agents, such as Paclitaxel and 5-FU, exert cytotoxicity toward not only cancer cells, but also normal cells, untoward effects are unavoidable. In contrast, JPH203, despite its higher IC50, would be expected to exert weaker adverse effects on normal cells because of its higher cancer cell selectivity. These pharmacological properties could redeem the high IC50 of JPH203. In 2023, a phase 2 clinical study also showed effective results at a similar concentration of JPH203 [34]. As the pharmacokinetics of this drug are inferred to be too complicated to be simply explained only based on blood concentrations, the detailed mechanism of pharmacokinetics is an issue to be addressed in future research.

We have added these points to the discussion (lines 475 to 494).

33. Oda, K.; Hosoda, N.; Endo, H.; Saito, K.; Tsujihara, K.; Yamamura, M.; Sakata, T.; Anzai, N.; Wempe, M.F.; Kanai, Y.; et al. l-Type amino acid transporter 1 inhibitors inhibit tumor cell growth. Cancer Sci. 2010, 101, 173-179.

34. Furuse, J.; Ikeda, M.; Ueno, M.; Furukawa, M.; Morizane, C.; Takehara, T.; Nishina, T.; Todaka, A.; Okano, N.; Hara, K.; et al. Nanvuranlat, an L-type amino acid transporter (LAT1) inhibitor for patients with pretreated advanced refractory biliary tract cancer (BTC): Primary endpoint results of a randomized, double-blind, placebo-controlled phase 2 study. J. Clin. Oncol. 2023, 41, 494-494.

If this inhibitor could target the stroma cells, would that mean the side effect of this inhibitor should be more carefully investigated?

[Response]

Thank you for your bringing this important topic up for discussion. The side effect on stromal cells is an issue that should be fully considered. As you pointed out, this LAT1 inhibitor attenuated stromal activation of colorectal tumors by suppressing cancer-associated fibroblasts (CAFs) and mesenchymal-epithelial transition (EMT) in cancer cells. However, our immunohistochemistry in clinical specimens showed that the expression of LAT1 in the normal mucosal epithelium and normal stromal cells is very low (Figure 1h). Therefore, we speculate that LAT1 may have a strong inhibitory effect on tumor stromal components such as cancer-associated fibroblasts and EMT of cancer cells, but the effect on normal stroma is relatively mild. We have added these points to the revised Discussion section (lines 531-536).

Since the metastasis tumor expressed more LAT1, would that be interesting to see the LAT1 inhibitor effect on controlling metastasis, using a more metastasis prone tumor model.

[Response]

We completely agree that we should evaluate the efficacy of the LAT1 inhibitor using a model with high metastatic potential. From that viewpoint, we focused on creating a tumor model with high metastatic potential in this study. We created the high metastatic model by co-implantation of mesenchymal stem cells with cancer cells and with the orthotopic implantation technique. We have previously verified that co-transplantation of MSCs or bone marrow-derived fibroblasts not only results in stromal abundance but also a high metastatic potential [1-3]. Furthermore, CT26, which we used in an in vivo experiment in this study, is a LAT1-high expressing cancer cell line (Figure 2a). In fact, in this study, 7 cases of lymph node metastasis were observed in 12 mice in the control group. We apologize for the lack of explanation on this point, and have added the relevant content to the Discussion section (lines 500-502).

20. Takigawa, H.; Kitadai, Y.; Shinagawa, K.; Yuge, R.; Higashi, Y.; Tanaka, S.; Yasui, W.; Chayama, K. Multikinase inhibitor regorafenib inhibits the growth and metastasis of colon cancer with abundant stroma. Cancer Sci. 2016, 107, 601-608.

21. Takigawa, H.; Kitadai, Y.; Shinagawa, K.; Yuge, R.; Higashi, Y.; Tanaka, S.; Yasui, W.; Chayama, K. Mesenchymal stem cells induce epithelial to mesenchymal transition in colon cancer cells through direct cell-to-cell contact. Neoplasia. 2017, 19, 429-438.

22. Yorita, N.; Yuge, R.; Takigawa, H.; Ono, A.; Kuwai, T.; Kuraoka, K.; Kitadai, Y.; Tanaka, S.; Chayama, K. Stromal reaction inhibitor and immune-checkpoint inhibitor combination therapy attenuates excluded-type colorectal cancer in a mouse model. Cancer Lett. 2021, 498, 111-120.

Minor:

The statistical method should be annotated in each figure legend.      

[Response]

Thank you for your useful advice. We have described the statistical analysis methods in the legends of Tables 1 and 2 and Figures 1, 2, and 5.

In figure 2a and 2c, if this difference reaches statistical significance?

[Response]

In the experiment shown in Figure 2a, PCR was conducted in triplicate. In the comparison with normal tissues, the difference in LAT1 expression was significant in most cells. We have added the information regarding the statistical analysis in Figure 2a and the figure legend.

In the experiments depicted in Figure 2c, the proliferation assay was conducted in quadruplicate. In comparison with that in control wells, proliferation was significantly inhibited above a certain concentration. The significant concentration differs, depending on cell lines. We have added information regarding the statistical analysis in the figure legend for Figure 2c.

In Figure 5c, it would be interesting to show the quantitative information from the image in each channel.

[Response]

Thank you for your invaluable comment. The positive area of vimentin compared with the E-cadherin-positive area is certainly interesting. We calculated the DAPI, vimentin, and E-cadherin staining area and have generated a bar graph to display the results (Figure 5d). The vimentin / DPAI-positive area rate was significantly lower in the JPH203 group than in the treatment group. The vimentin / E-cadherin-positive area rate was significantly lower in the JPH203 group than in the treatment group.

We have also added these findings to the Results section (lines 391-393).

Reviewer 2 Report

The aim of this original work was to „elucidate the relationship between large neutral amino acid transporter 1 (LAT1) expression and tumor progression and investigate the anti-tumor effects and mechanism of LAT1 inhibition in CRC”.

The objectives and the introduction of the thesis are written clearly, I have no comments on this part of the thesis. Although the hypotheses of the thesis are expressed, I would recommend clearly writing the aim of the thesis in the introduction of the thesis and in the abstract. Better is this I sentence from the Discussion section to state the purpose of the work than the one attached in the Introduction.

I would also recommend shortening and rewording the title of the paper a bit and reducing the number of papers cited (there are 50, too many for an original paper).

Material and Methods: the research is based on all modern in vitro and in vivo (female BALB/c mice, 154 consecutive patients with CRC) techniques used today, such as cell culture using different panel of colorectal cells and control cell line.

The description of molecular, IHC, bioinformatics and statistical techniques is correct. Here I have only few comments.

In the description of Material and Methods, please complete the abbreviation of the name MSCs used I time, because it is missing. Also, in line 250, please add the word "software" after EZR or other appropriate words. In line 95-96, the sentence "in normal human tissues and tumors as well as in normal colorectal mucosa and colorectal tumor tissues" is incomprehensible...throughout the paper, please make the spelling "H.E." more correct. to "H&E" or "H+E" staining

The results are presented accurately, although some Figures (e.g. Figure 2a (p. 10); Figure 3 - twice the word "control" , truncated part of the text in the heading), imaging photographs from under the microscope with the IHC or H&E reaction in this version of the paper, are unacceptable (too small and illegible, Fig. 1h, Fig. 5a and 5b). In Fig. 4b - please correct "Amino sugar" to "Amino acid" metabolism; and Fig. 4d - "T cell" also seems inappropriate; rather "Immune response" as above (Fig. 4b)?

Section 3.2 - I recommend deleting the re-description of the semi-quantitative scale for assessing the IHC reaction and going straight to the result.

The discussion is nicely written, I just don't understand the sentence in lines 427-428, indicating that LAT1 is expressed everywhere in high amounts even in "normal tissues", please verify this.

The statement "the utility of JPH203" seems to lack the completion of the sentence, i.e. „the utility of this protein to assess tumor progression, to evaluate prognosis or diagnosis, etc. please think (also in the title of the paper).

References: only 13 of the 50 papers are from the last 5 years (26%), please discard these especially from those cited in the Materials and Methods section, which can be removed.

To sum up:

1.      It should be completed the aim of the study (in Introduction and abstract).

2.      Please consider shortening the title of your paper and reducing the number of works cited.

3.      Some minor editorial errors in the Figure in chapter Results.

4.      The paper can be published after completion of minor comments, small linguistic correction and presenting a better quality of figures.

Author Response

[Response]

We are grateful for the positive review of our manuscript and for the constructive comments and useful suggestions.

We have revised the manuscript in response to the reviewers' comments and had our manuscript checked by the English Editing Service Editage (https://www.editage.jp). We have provided our point-by-point responses to the reviewers’ comments below. Our responses are indicated in red font. Revisions to the manuscript are also indicated in red font.

The aim of this original work was to „elucidate the relationship between large neutral amino acid transporter 1 (LAT1) expression and tumor progression and investigate the anti-tumor effects and mechanism of LAT1 inhibition in CRC”.

The objectives and the introduction of the thesis are written clearly, I have no comments on this part of the thesis. Although the hypotheses of the thesis are expressed, I would recommend clearly writing the aim of the thesis in the introduction of the thesis and in the abstract. Better is this I sentence from the Discussion section to state the purpose of the work than the one attached in the Introduction.

[Response]

Thank you for your helpful advice. We have stated the aim of this study in the last part of the Introduction (lines 88-90), per your instruction.

I would also recommend shortening and rewording the title of the paper a bit and reducing the number of papers cited (there are 50, too many for an original paper).

[Response]

We agree with your observation that there were too many references and the title was too long. We have thoroughly reduced and revised the references. We have also changed the title of our manuscript to “The anti-tumor effect of the newly developed LAT1 inhibitor JPH203 in colorectal carcinoma, according to a comprehensive analysis”.

Material and Methods: the research is based on all modern in vitro and in vivo (female BALB/c mice, 154 consecutive patients with CRC) techniques used today, such as cell culture using different panel of colorectal cells and control cell line.

The description of molecular, IHC, bioinformatics and statistical techniques is correct. Here I have only few comments.

In the description of Material and Methods, please complete the abbreviation of the name MSCs used I time, because it is missing. Also, in line 250, please add the word "software" after EZR or other appropriate words. In line 95-96, the sentence "in normal human tissues and tumors as well as in normal colorectal mucosa and colorectal tumor tissues" is incomprehensible...throughout the paper, please make the spelling "H.E." more correct. to "H&E" or "H+E" staining

[Response]

The full expression has been provided for MSCs in the simple summary section (line 30) and in the main text on line 148.

We also have added “software” after EZR on line 251. 

We have revised the sentence "in normal human tissues and tumors as well as in normal colorectal mucosa and colorectal tumor tissues" to read “in normal tissue and tumor tissues obtained from colon and other organs.” We apologize for the confusion.

We have also changed the term “H.E.” into “H & E” throughout the document. Thank you.

The results are presented accurately, although some Figures (e.g. Figure 2a (p. 10); Figure 3 - twice the word "control", truncated part of the text in the heading), imaging photographs from under the microscope with the IHC or H&E reaction in this version of the paper, are unacceptable (too small and illegible, Fig. 1h, Fig. 5a and 5b). In Fig. 4b - please correct "Amino sugar" to "Amino acid" metabolism; and Fig. 4d - "T cell" also seems inappropriate; rather "Immune response" as above (Fig. 4b)?

[Response]

We have thoroughly revised the figures and have now provided all figures in high-resolution. We have also enlarged the IHC and H & E figures.

We have changed Amino sugar to Amino acid metabolism, per your instruction.

As you pointed out, T cell is not appropriate; “Immune response” is correct. We have revised it (Figure 4d).

Section 3.2 - I recommend deleting the re-description of the semi-quantitative scale for assessing the IHC reaction and going straight to the result.

[Response]

Per your recommendation, we have deleted the re-description for the semi-quantitative scale and revised the sentence in the Results section 3.2 (lines 287-288). Thank you for helping us improve the readability of our document.

The discussion is nicely written, I just don't understand the sentence in lines 427-428, indicating that LAT1 is expressed everywhere in high amounts even in "normal tissues", please verify this.

[Response]

We apologize for our careless mistake here. Thank you for identifying this critical misdescription. What we intended to explain here was that LAT1 expression differed depending on the tumor stage as follows:

Metastases > recurrent tumors > primary tumors > normal tissues.

As you pointed out, LAT1 expression in normal tissue was very low. We have revised the sentences, as follows.

“The expression of LAT1 was higher in the following order: metastases, recurrent tumors, primary tumors, and normal tissues. This result suggests that it is a molecule involved in tumor progression and metastasis” (lines 447-450).

We have also revised the results 3.1 section (line 260) to align with this revision.

The statement "the utility of JPH203" seems to lack the completion of the sentence, i.e. „the utility of this protein to assess tumor progression, to evaluate prognosis or diagnosis, etc. please think (also in the title of the paper).

[Response]

Thank you for your kind suggestion. We have revised the sentence as follows;

“In this study, we demonstrated the utility and mechanism of JPH203, a newly developed LAT1-specific inhibitor, to inhibit tumor progression using a unique stroma-abundant allogeneic immunoreactive mouse colorectal tumor created by orthotopic transplantation of a mouse-derived CRC cell line and MSCs in the mouse cecum” (lines 537-540).

We have also changed the title of our manuscript to “The anti-tumor effect of the newly developed LAT1 inhibitor JPH203 in colorectal carcinoma, according to a comprehensive analysis.”

References: only 13 of the 50 papers are from the last 5 years (26%), please discard these especially from those cited in the Materials and Methods section, which can be removed.

[Response]

Thank you for your careful review. We have thoroughly revised the references and reduced the number to 40, specifically reducing the number of older papers in the Material and Methods section.

To sum up:

  1. It should be completed the aim of the study (in Introduction and abstract).
  2. Please consider shortening the title of your paper and reducing the number of works cited.
  3. Some minor editorial errors in the Figure in chapter Results.
  4. The paper can be published after completion of minor comments, small linguistic correction and presenting a better quality of figures.

[Response]

We have thoroughly revised our manuscript per your individual instructions.

Thank you again.

Reviewer 3 Report

In this study, the authors explored the antitumor effects and mechanism of LAT1 inhibition in CRC and explained the link between LAT1 expression and tumor growth. In colorectal tissues, LAT1 expression was greater than in normal tissues. Patients with significant LAT1 expression had a bad prognosis since LAT1 expression increased as the tumor progressed. Immunostaining of CRC specimens found that 68.8% of patients exhibited dominant LAT1 expression at the location of the tumor. LAT1 was also discovered to be substantially expressed in the majority of CRC cell lines relative to normal mucosa. In a mouse model, JPH203 successfully suppressed the proliferation of CRC cell lines and tumor growth and metastasis. A comprehensive investigation demonstrated that injection of a LAT1 inhibitor inhibits not just proliferative and amino acid metabolic pathways but also stroma-related processes.

This is a very well-prepared, well-designed, high-quality study. In vivo experiments using in silico results have been used to present multiple validated results. The figures and tables are illustrative.
Their results are understandable and clear.
The results are presented in light of existing knowledge and are logically integrated to generate new directions for both research and therapy. They also explain the limitations of the study in a clear way.
In the diagram in Figure 2, I suggest correcting the final part of the diagram in addition to the SD values, as the caption there are incorrect and breaks.
Apart from this, I recommend that the paper be accepted for publication.

Author Response

[Response]

We are grateful for the positive review of our manuscript and for the constructive comments and useful suggestions.

We have revised the manuscript in response to the reviewers' comments and had our manuscript checked by the English Editing Service Editage (https://www.editage.jp). We have provided our point-by-point responses to the reviewers’ comments below. Our responses are indicated in red font. Revisions to the manuscript are also indicated in red font.

In this study, the authors explored the antitumor effects and mechanism of LAT1 inhibition in CRC and explained the link between LAT1 expression and tumor growth. In colorectal tissues, LAT1 expression was greater than in normal tissues. Patients with significant LAT1 expression had a bad prognosis since LAT1 expression increased as the tumor progressed. Immunostaining of CRC specimens found that 68.8% of patients exhibited dominant LAT1 expression at the location of the tumor. LAT1 was also discovered to be substantially expressed in the majority of CRC cell lines relative to normal mucosa. In a mouse model, JPH203 successfully suppressed the proliferation of CRC cell lines and tumor growth and metastasis. A comprehensive investigation demonstrated that injection of a LAT1 inhibitor inhibits not just proliferative and amino acid metabolic pathways but also stroma-related processes.

This is a very well-prepared, well-designed, high-quality study. In vivo experiments using in silico results have been used to present multiple validated results. The figures and tables are illustrative.
Their results are understandable and clear.
The results are presented in light of existing knowledge and are logically integrated to generate new directions for both research and therapy. They also explain the limitations of the study in a clear way.

[Response]

We greatly appreciate your positive review and kind comments.

In the diagram in Figure 2, I suggest correcting the final part of the diagram in addition to the SD values, as the caption there are incorrect and breaks.
Apart from this, I recommend that the paper be accepted for publication.

[Response]

Thank you for your comment. We apologize for these careless mistakes. As you pointed out, the caption for Figure 2A was improperly presented. We have revised the caption for Figure 2A and added the SDs with error bars in the bar graphs.

Round 2

Reviewer 1 Report

Authors addressed all my questions.